# Flavonoids and Sesquiterpene Lactones from *Lychnophora ericoides* (Arnica-Do-Cerrado) and Their In Vitro Effects on Multiple Myeloma and Acute Myeloid Leukemia

**DOI:** 10.3390/metabo15080542

**Published:** 2025-08-09

**Authors:** Calisto Moreno Cardenas, Ren Ove Kratzert, Sofie Hanifle, Elida Cleyse Gomes da Mata Kanzaki, Isamu Kanzaki, Brigitte Kircher, Serhat Sezai Çiçek

**Affiliations:** 1Department of Pharmaceutical Biology, Kiel University, Gutenbergstraße 76, 24118 Kiel, Germany; ccardenas@pharmazie.uni-kiel.de (C.M.C.);; 2Department of Biotechnology, Hamburg University of Applied Sciences, Ulmenliet 20, 21033 Hamburg, Germany; 3Department of Internal Medicine V (Hematology and Oncology), Medical University of Innsbruck, Anichstraße 35, 6020 Innsbruck, Austriabrigitte.kircher@i-med.ac.at (B.K.); 4Tyrolean Cancer Research Institute, Innrain 66, 6020 Innsbruck, Austria; 5Laboratory of Bioprospection, Darcy Ribeiro University Campus, University of Brasilia, Brasilia 70910-900, DF, Brazilcalipnus@protonmail.com (I.K.)

**Keywords:** *Lychnophora pinaster*, arnica da serra, Cichorioideae, terpene, polyphenol, structure-activity-relationship, cytotoxicity, metabolic activity, hematologic cancer, tumor

## Abstract

Objectives: Multiple myeloma and acute myeloid leukemia are severe forms of blood cancer, which lack effective therapies for treatment. In our search for new chemical lead structures from nature, we were investigating the Brazilian medicinal plant arnica-do-cerrado (*Lychnophora ericoides*). Methods: Repeated chromatography led to the isolation of four flavonoids and three sesquiterpenoids, which were evaluated for their cytostatic and cytotoxic properties against HL-60, MOLM-13, AMO-1, and KMS-12 PE cancer cells as well as the non-malignant HS-5 cell line. Results: Whereas the isolated flavonoids displayed only moderate activity, the three sesquiterpene lactones goyazensolide, centratherin, and lychnopholide exhibited pronounced effects against all four tested cell lines. Goyazensolide was the most effective compound, inhibiting proliferation and metabolic activity with IC_50_ values between 1.0 and 1.6 µM, as well as 1.0 to 2.0 µM, respectively. Centratherin and lychnopholide were somewhat less active but showed higher selectivity towards malignant cell lines, which was most pronounced for MOLM-13 cells. Conclusion: The results of this study revealed interesting natural products that will be further evaluated for their potential as new lead compounds for the treatment of acute myeloid leukemia and multiple myeloma.

## 1. Introduction

Multiple myeloma (MM) and acute myeloid leukemia (AML) are very heterogeneous forms of cancer that proliferate extensively within the bone marrow and account for more than 10% of all blood cancers [1,2,3]. Showing a variety of genetic variations, resistant clones are selected upon MM treatment, thus hampering effective therapies and often leading to disease relapses [2]. In the same way, AML displays multiple genetic abnormalities occurring in myeloid precursor cells that cause most patients to succumb to the disease [3]. Therefore, novel therapies are urgently warranted. Natural products are a potent source for effective therapeutics, either constituting the active drug themselves or serving as lead structures for the development of new synthetic compounds [4]. Especially in cancer treatment, drug discovery from nature is still the most common means of the development of novel therapeutics. In our ongoing research on new natural leads, which has yielded both flavonoids [5,6,7] and terpenoids [8,9] with MM and/or AML inhibitory activity thus far, we now focus on the isolation of cytotoxic principles from arnica-do-cerrado (*Lychnophora ericoides*) and their effects on HL-60 and MOLM-13 (AML) as well as AMO-1 and KMS-12 PE (MM) cell lines.

*Lychnophora ericoides* Mart. (Asteraceae, Cichorioideae), formerly known as *L. pinaster* [10], is a species widely used in traditional Brazilian medicine for the treatment of pain, inflammation, and rheumatism. The genus *Lychnophora* is predominantly distributed in the coastal states of Bahia, Goias, and Minas Gerais, and contains characteristic sesquiterpenoids, such as germacranolides and furane heliangolides [11,12,13,14,15]. Previous studies showed that extracts of *Lychnophora* species exhibit analgesic [16,17], anti-inflammatory [18], and trypanocidal effects [19,20,21], indicating the pharmaceutical potential of these species. Regarding our investigation, the mentioned sesquiterpenoids were of particular interest, as their potential for use against MM was mentioned in several studies [22], as well as eventual methylated flavonoids, which were recently discovered to show pronounced in vitro activity against AML [7]. As a result, eight compounds were isolated from the aerial parts of arnica-do-cerrado, of which seven (**1**–**7**) were subjected to biological evaluation (Figure 1).

## 2. Materials and Methods

### 2.1. Plant Material, Reagents and Experimental Procedures

Dried aerial parts (leaves and flowers) of *L. ericoides* were collected in January 2023 at the Fazenda Touro Bravo in Alto Paraíso de Goiás, Brazil, at an altitude of 874 m above sea level. The coordinates were 53.847′13″ S/14.445′047″ W. A voucher specimen was deposited under no. 23012023LB at the Laboratory of Bioprospection, University of Brasília. The species was registered for access to the genetic heritage in SisGen with process no. 00.038.174/0001-43. LC-MS-grade acetonitrile and water, as well as other (analytical grade) solvents and reagents, were purchased from VWR International GmbH (Darmstadt, Germany). LC-MS-grade formic acid was purchased from Sigma–Aldrich Co. (St. Louis, MO, USA). The water used in the semi-preparative HPLC was double-distilled in-house. DMSO-*d*_6_ (99.80%, Lot T2831, Batch 0920-2D) and MeOH-*d*_4_ (99.80%, Lot T2091, Batch 0720) for NMR spectroscopy were purchased from Euriso-top GmbH (Saarbrücken, Germany). TLC was performed on silica gel 60 F254 plates (VWR International, Darmstadt, Germany) using hexane-ethyl acetate–methanol (15:10:2) as the mobile phase and vanillin–sulfuric acid as the detection reagent. Flash chromatography was performed on a Büchi Pure C-810 Flash chromatograph using a FlashPure Ecoflex cartridge (silica 50 µm irregular, 120 g, Büchi Labortechnik GmbH, Essen, Germany). Column chromatography was performed using a Sephadex LH-20 column (GE Healthcare AB, Uppsala, Sweden). Semi-preparative HPLC was performed using an Ultimate 3000 instrument equipped with an HPG-3400SD pump, WPS-3000SL autosampler, TCC-3000SD column heater, and VWD-3400RS variable wavelength detector (Thermo Fisher Scientific Inc., Waltham, MA, USA) using a Phenomenex Aqua column (5 µm, 250 mm × 10 mm). Extracts, fractions, and pure compounds were analyzed on a Shimadzu Nexera 2 liquid chromatograph connected to an LC-MS triple quadrupole mass spectrometer with electrospray ionization (ESIMS, Shimadzu, Kyoto, Japan). For separation, a Phenomenex Luna Omega polar C18 column (100 mm × 2.1 mm, 1.6 µm particle size, Phenomenex, Aschaffenburg, Germany) was used. 1D (^1^H and ^13^C) and 2D (HSQC, HMBC, and COSY) NMR spectra were recorded on a Bruker Avance III 400 NMR spectrometer operating at 400 MHz for the proton channel and 100 MHZ for the ^13^C channel with a 5 mm PABBO broadband probe with a z-gradient unit at 298 K (Bruker BioSpin GmbH, Rheinstetten, Germany). Reference values were 2.50 (^1^H) and 39.51 (^13^C) for dimethyl sulfoxide and 3.31 (^1^H) and 49.15 (^13^C) for methanol. Structure elucidation was performed on Topspin 3.6 software (Bruker Biospin GmbH, Rheinstetten, Germany). NMR sample tubes of 5 mm were obtained from Rototec-Spintec GmbH, Griesheim, Germany.

### 2.2. Extraction and Isolation

The powdered plant material (992 g) was divided into two flasks and, after ultrasonication for 15 min, extracted through maceration at room temperature for a total of twelve days. The solvents used were *n*-hexane (15.5 L, 3 days), acetone (21.8 L, 5 days), and methanol (18.6 L, 4 days), yielding 46.4, 59.7 and 57.35 g of crude extracts, respectively. The acetone extract (59.7 g) was further processed by means of vacuum liquid chromatography (VLC) and subsequent elution with 6.55 L of *n*-hexane, 3.0 L of *n*-hexane–dichloromethane (1:1), 2.0 L of dichloromethane, 2.0 L of dichloromethane–acetone (1:1), 2.2 L of acetone, 2.0 L of acetone–methanol (1:1), and 2.0 L of methanol, yielding seven fractions (A to G). Chromatographic analysis of fractions A and B showed UV and mass spectra that hinted at the compounds of our interest and were therefore further processed. Fraction A (*n*-hexane, 20.92 g) was subjected to flash chromatography with a gradient consisting of *n*-hexane, ethyl acetate, and MeOH, with isocratic elution for 5 min with 100% *n*-hexane, and gradient elution for 60 min to 100% ethyl acetate and for 30 min to 100% methanol, followed by isocratic elution for 5 min with 100% methanol, leading to eight fractions (A1 to A8). Fraction A5 was further separated using Sephadex LH-20 column chromatography (2 × 100 cm) in a dichloromethane–acetone (85:15) mixture, yielding eight fractions (A5A to A5H), of which fraction A5D consisted of 28 mg of compound **8**. Fractions A5A and A5B were combined and further purified by semi-preparative HPLC using isocratic elution for 30 min with 50% acetonitrile at a flowrate of 2 mL/min and 35 °C oven temperature yielding 8 mg of compound **7** and **5** mg of compound **1**. The VLC fraction B (dichloromethane, 3.29 g) was also separated using Sephadex LH-20 column chromatography, resulting in 50 fractions (B1 to B50), of which fraction B32 afforded 22 mg of compound **2** and fraction B48 afforded 26 mg of compound **3**. Fractions B10 and B43 were further purified by semi-preparative HPLC. For fraction B10, a gradient of acetonitrile and water with 25% to 55% acetonitrile in 20 min at a flowrate of 3 mL/min and an oven temperature of 35 °C was applied, resulting in 10 mg of compound **5** and 21 mg of compound **6**. The separation of fraction B43 was achieved using isocratic elution with 70% acetonitrile at a flowrate of 2 mL/min at 35 °C oven temperature, yielding 31 mg of compound **4**. The purity of the tested compounds was determined using the built-in ERETIC2 feature of Bruker Topspin. For the isolated flavonoids, purities of 82.8% (**1**), 91.7% (**2**), 64.3% (**3**), and 97.8% (**4**) were calculated, while the sesquiterpenoids showed purities of 99.0% (**5**), 99.4% (**6**), and 97.0% (**7**), respectively. For the evaluation of biological activity, 50 mM stock solutions of the lyophilized compounds were prepared and stored at 4 °C.

### 2.3. Analysis of Proliferation and Metabolic Activity

Acute myeloid leukemia cell line HL-60 and multiple myeloma cell lines AMO-1 and KMS-12-PE were obtained from the German Collection of Microorganisms and Cell Cultures (DSMZ), Braunschweig, Germany, and non-tumorigenic stromal cell line HS-5 was acquired from the American Type Culture Collection (ATCC, Manassas, VA, USA). All cell lines were cultured in RPMI 1640 medium without phenol red (Pan Biotech, Aidenbach, Germany), supplemented with 2 mM glutamine, 100 U/mL penicillin, 100 µg/mL streptomycin (Sigma-Aldrich, St. Louis, MO, USA), and 10% fetal bovine serum (FBS Maximus, Catus Biotech GmbH, Tutzing, Germany) at 37 °C in a humidified atmosphere of 5% CO_2_/95% air. Cells were fed twice weekly. Logarithmically growing cells were resuspended at 1 × 10^5^ cells/mL, plated in triplicate in round-bottom 96-well plates (100 µL per well; Falcon, Becton Dickinson, Franklin Lakes, NJ, USA), and incubated at 37 °C in 5% CO_2_/95% air for 2 h. Subsequently, the compounds were adjusted in the cell culture medium to reach the final concentrations in a total volume of 150 µL and incubated for an additional 72 h. Cell proliferation was assessed by [^3^H]-thymidine uptake. Each well was exposed to 2 µCi [^3^H]-thymidine (Hartmann Analytic, Braunschweig, Germany) during the final 12–16 h of incubation. Cells were harvested using a semi-automated device, and [^3^H]-thymidine incorporation (expressed in counts per minute, cpm) was measured with a scintillation counter (MicroBeta TriLux, PerkinElmer, Waltham, MA, USA). Metabolic activity was assessed by resazurin staining. Resazurin (Carl Roth, Karlsruhe, Germany) was dissolved in distilled water at a concentration of 0.1 mg/mL, and 15 µL of this solution was added to each well, followed by a 2 h incubation at 37 °C. Relative fluorescence units (RFUs) were measured using a Synergy H1 plate reader equipped with a 560 nm excitation/590 nm emission filter set (BioTek; Agilent, Santa Clara, CA, USA). Proliferation and metabolic activity in the absence of compounds were defined as 100% (control), and the values for compound-treated samples were expressed as percentages relative to the control. Selectivity indices were calculated by dividing the IC_50_ value of each compound against HS-5 cells through the IC_50_ values against the investigated cell lines.

### 2.4. Statistical Analysis

The Wilcoxon Rank Sum Test was used to analyze proliferation and metabolic activity in the absence and presence of varying compound concentrations (NCSS 2007 software, version 07.1.2, Kaysville, UT, USA). A *p*-value < 0.05 was considered statistically significant. IC_50_ values were calculated using GraphPad Prism 10.1.2. This software reports 95% confidence intervals instead of standard errors. In some cases, however, a 95% confidence interval could not be determined; these data points are indicated as “not determined” (n.d.). Nonetheless, the figures clearly illustrate the IC_50_ values due to the narrow concentration range tested.

## 3. Results

### 3.1. Isolation and Identification

Exhaustive maceration followed by vacuum liquid chromatography (VLC) and repeated chromatographic separation led to the isolation of three flavonols (**1**, **2**, and **4**), one flavone (**3**), three sesquiterpene lactones (**5**, **6**, and **7**) and one triterpene (**8**) (Figure 1). After a comparison of mass spectrometry (MS) and nuclear magnetic resonance (NMR) data with the literature reports, the isolated compounds were identified as retusin (**1**) [23], 7,3′,4′-trimetoxymyricetin (**2**) [24], diosmetin (**3**) [25], 3,7-di-O-methylquercetin (**4**) [26], goyazensolide (**5**) [27], centratherin (**6**) [28], lychnopholide (**7**) [29], and lupeol (**8**) [30]. MS and NMR data of all isolated compounds are provided in Appendix A.

### 3.2. Antiproliferative Activity

The isolated flavonoids (**1**–**4**) and sesquiterpenoids (**5**–**7**) were initially tested at various concentrations for their effects on the proliferation of AML (HL-60 and MOLM-13) and myeloma (AMO-1 and KMS-12 PE) cell lines (Table A1, Figure 2). Compound **8** was not tested because of its very low polarity, hampering its solubility in aqueous media. All tested compounds dose-dependently exhibited activity against AML and MM cells; however, there were notable differences between the two compound classes. The flavonoid retusin (**1**) only partially inhibited proliferation at concentrations of 50 and 100 µM, whereas 7,3′,4′-trimethoxymyricetin (**2**) significantly suppressed proliferation at 100 µM and remained effective at 50 µM (Table A1). In contrast, diosmetin (**3**) was the only flavonoid to display differential activity, showing only weak antiproliferative effects against HL-60 cells but moderate effects on MOLM-13 cell proliferation. The strongest antiproliferative activity on AML cells was observed for 3,7-di-O-methylquercetin (**4**), which significantly inhibited proliferation by more than 50% in both cell lines at a concentration of 10 µM. Effects on the two myeloma cell lines AMO-1 and KMS-12 PE were in the same range. 7,3′,4′-trimethoxymyricetin (**2**) exhibited somewhat stronger effects than those against AML cells, whereas 3,7-di-O-methylquercetin (**4**) demonstrated lower activity. Moreover, the effects of diosmetin (**3**) against both myeloma cell lines were more pronounced than on HL-60 cells but less potent compared to its inhibition of the MOLM-13 cell line.

In contrast to the investigated flavonoids, all three sesquiterpenoids completely inhibited the proliferation of both HL-60 and MOLM-13 cells at a concentration as low as 5 µM (Figure 2a,b). Goyazensolide (**5**), centratherin (**6**), and lychnopholide (**7**) reduced cell proliferation in a dose-dependent manner, yielding IC_50_ values of 1.1, 1.2, and 1.3 µM, respectively, against HL-60 cells (Table 1). A similar pattern was observed for the MOLM-13 cell line, with 50% proliferation inhibition occurring at concentrations of 1.0, 1.1, and 1.4 µM, respectively. Although all three sesquiterpenoids inhibited the proliferation of myeloma cells in a dose-dependent manner (Figure 2c,d), these cell lines were less sensitive to treatment compared to AML cells. Again, goyazensolide (**5**) exhibited slightly stronger effects than the other two sesquiterpenoids, showing 50% inhibition of proliferation at concentrations of 1.6 µM in AMO-1 and 1.5 µM in KMS-12-PE cells (Table 1). In contrast, compounds **6** and **7** reached similar levels of inhibition at concentrations of 2.2 and 1.8 µM in AMO-1 cells, and 2.2 and 2.6 µM in KMS-12-PE cells, respectively. The concentrations needed to inhibit 50% of the proliferating cells seem to be higher than those in drugs used for the treatment of myeloma. We have shown that doxorubicin inhibited the proliferation of KMS-12-PE at a concentration of 10 nM to 45.5% [31]. The immunomodulatory lenalidomide was very potent in inhibiting the proliferation of KMS-12-BM, the sister cell line of KMS-12-PE, (IC_50_ = 57 nM), but other MM cell lines required IC_50_ concentrations in the high µM range to induce similar effects [32].

Finally, the effects of the three sesquiterpenoids on the proliferation of the non-malignant HS-5 stroma cell line were investigated (Figure 3). This cell line was chosen because stroma cells play an important role in the bone marrow niche of MM patients [33] and are considered to protect AML cells from therapies [34]. At lower concentrations (1–2 µM), the compounds did not reduce the proliferation of HS-5 cells as strongly as in the leukemia cell lines. Compound **7** exhibited significantly weaker effects against the non-malignant cell line, resulting in selectivity indices of more than 3 against both AML cell lines and still 2.0 and 1.7 against AMO-1 and KMS-12 PE (Table 1).

### 3.3. Metabolic Activity

Proliferation inhibition often leads to cell death. Therefore, the cytotoxic activity of the compounds was further evaluated using a modified 3-(4,5-dimethylthiazol-2-yl)-2,5-diphenyltetrazolium bromide (MTT) assay, the resazurin (Alamar Blue) assay. Metabolically active cells reduce the non-fluorescent dye resazurin to the fluorescent dye resorufin via mitochondrial reductase activity, which correlates with cell viability. In the first step the metabolic activity of the four isolated flavonoids was investigated, showing a dose-dependent reduction, but to a much lower extent than in the proliferation assays (Table A2). In fact, only 7,3′,4′-trimethoxymyricetin (**2**) inhibited the metabolic activity of all four cell lines by more than 50% at a concentration of 100 µM. Additionally, diosmetin (**3**) and 3,7-di-O-methylquercetin (**4**) induced antimetabolic activity against the MOLM-13 or the AMO-1 cell line at the same concentration.

The three sesquiterpenoids reduced the metabolic activity of all four cell lines in a dose-dependent manner, with the strongest effects again being observed in MOLM-13 cells (Figure 4), resulting in IC_50_ values of 1.0 µM (**5**), 1.3 µM (**6**), and 1.8 µM (**7**) (Table 2). All three compounds also had a significant impact on the other investigated AML cell line (HL-60), with IC_50_ values of 1.7 µM (**5**), 1.9 µM (**6**), and 2.9 µM (**7**). As observed in the proliferation assays, myeloma cells were somewhat less affected by the three sesquiterpenoids, with IC_50_ values ranging from 2.0 to 3.4 µM against the AMO-1 cell line and from 1.9 to 4.0 µM against KMS-12 PE cells, respectively. These IC_50_ values are clearly higher in comparison to the proteasome inhibitor bortezomib [35,36], an agent which has dramatically improved the prognosis of MM patients [37]. However, overall, the cytotoxic activity of the compounds was less pronounced than their cytostatic effects, as was also observed for doxorubicin [31]. However, regarding the metabolic activity, the inhibition of non-malignant HS-5 cells by compounds **5**–**7** was considerably lower. This effect was most evident for the MOLM-13 cell line, with SI values of 2.6 (**5**), 3.5 (**6**), and 2.8 (**7**) (Figure 5, Table 2) indicating the higher selectivity of the compounds against AML than MM cells. Similar SI values can be calculated from the results published by Schmidmaier et al., who treated HS-5 cells with drugs used for the treatment of MM and AML [38]. Cisplatin must be used at concentrations above 1 µM to inhibit HS-5 cells [39].

## 4. Discussion

Flavonoids are widespread in the plant kingdom and, as non-caloric accompanying substances, also have nutritional value [22]. Alongside the regular consumption of fruits and vegetables, flavonoids might as well play a role in the prevention of tumor development [40]. Here, (poly)methylated flavonoids have become of particular interest because of their substantial chemopreventive effects, and their higher metabolic stability compared to their unmethylated counterparts, enabling them to circulate longer in the body [41,42,43,44]. In our previous study, we identified anti-cancer activity in the two methylated flavonoids cirsimaritin and xanthomicrol, which exhibited pronounced inhibitory effects at low micromolar concentrations in a modified MTT assay. This method is comparable to the resazurin-based assay in the present study. Furthermore, we discovered FLT3 inhibition to be a mode of action of these two compounds [7]. With the newly isolated compounds (**1** to **4**) showing different methylation patterns, the structure–activity-relationship was further examined. Although two of the four tested flavonoids showed lower purities (thus hampering proper conclusions regarding their structure–activity relationship) it can still be seen that only one compound (**4**) induced broad antiproliferative effects in the low-micromolar range. This confirms that free hydroxy group in position 4′ is an important feature, which was a finding of our previous study. However, the effects of compound **4** did not reach those of cirsimaritin and xanthomicrol, which showed additional methoxylation in ring A and no substitution in positions 3 and 3′. The latter fact also confirms the hindrance of further substitutions in the B-ring, at least with respect to FLT3 inhibition.

Sesquiterpene lactones are most often found in the Asteraceae family, where they act as defensive compounds against herbivores and microorganisms and are responsible for the bitter taste of some sorts of lettuce [45]. They have been the subject of numerous publications on their anti-cancer activities and belong to the group of highly active natural substances. Artemisinin, thapsigargin, and parthenolide are prominent examples of the sesquiterpene lactones that were introduced into clinical trials [46]. Their anti-cancer activities range from influencing the sarcoplasmic reticulum Ca^2+^ ATPases (SERCA) via the inhibition of NF-κB, angiogenesis, metastasis, and the transcription factor STAT3 to the activation of the p53 signaling pathway and formation of reactive oxygen species (ROS) [46,47,48,49]. Despite their pronounced bioactivities, sesquiterpene lactones are generally considered unstable compounds due to their cyclic ester function. Also, their low water solubility and consequently decreased bioavailability, is an issue which must be addressed during drug development. However, with the technological advances that have been made in the formulation of drugs (e.g., nanospheres), many of these limitations can be handled. This may be one of the reasons why natural products are still the most common source for the development of new therapeutics, a fact that is especially evident in cancer therapy. Still, like many other drugs, natural products do show undesirable side effects when administered in higher doses. Such examples include the dose-dependent neurotoxicity of artemisinin or the tendency of sesquiterpene lactones to cause allergic reactions [50,51]. Chemical modification during drug development or eventual formulation measures (e.g., drug targeting) may also be an option.

Of the sesquiterpene lactones investigated in our study, goyazensolide (**5**) was identified as a potent NF-κB inhibitor in the human colon cancer cell line HT-29, suggesting that the inhibition of NF-κB expression may sensitize HT-29 to apoptosis [52]. Dos Santos et al. evaluated the effect of compound **5** against leukemia, breast, colon, murine skin, prostate, and central nervous system cancer cells, resulting in IC_50_ values between 0.17 (CEM) and 2.08 µM (B16) [12]. Goyazensolide (**5**) and lychnopholide (**7**) showed promising activities against human colon cancer cell line HT-29, with IC_50_ values of 0.56 and 1.4 µM, respectively, but were also found to inhibit the NF-κB p65 at concentrations of 3.8 and 2.9 µM [29]. At a later stage, Ren et al. observed the inhibition of AML (MOLM-13 and EOL-1) and HT-29 cells by compound **5**, with IC_50_ values of 0.86, 1.1 and 0.56 µM, respectively [53], which was confirmed by this study and our former results [7]. Furthermore, in a study on the inhibition of c-Myb-dependent gene expression, goyazensolid (**5**) showed the highest activity of all tested compounds, with an IC_50_ of 0.63 µM [54]; however, cytotoxic activity against the myelomonocytic chicken cell line HD11-C3 revealed an IC_50_ of 10.05 µM. Vongvanich et al. tested goyazensolid (**5**) and centratherin (**6**) against NCI-H187, KB and BC cell lines, with the two furanoheliangolides showing IC_50_ values of 0.014 (0.04 µg/mL) and 0.026 µM (0.07 µg/mL), respectively, against small lung cancer cell line NCI-H187 and IC_50_ values below 0.3 µM against the other two cell lines [55]. Centratherin (**6**) was also tested against the two glioma cell lines U251 and U87-MG, with IC_50_ values of 3.02 (8.06 µg/mL) and 1.34 µM (3.57 µg/mL), respectively [56]. Additionally, Machado et al. observed that centratherin (**6**) induced reactive oxygen species, resulting in high cytotoxic activity (IC_50_ from 0.48 to 1.31 µM) against nine tumor cell lines [28].

In our study, all three sesquiterpenoids—goyazensolide (**5**), centratherin (**6**), and lychnopholide (**7**)—exhibited pronounced effects against the two AML cell lines HL-60 and MOLM-13, as well as against the myeloma cell lines AMO-1 and KMS-12 PE. The compounds showed activity in the low-micromolar range, with IC_50_ values ranging from 1.0 to 2.6 µM for the inhibition of proliferation and from 1.0 and 4.0 µM for the inhibition of metabolic activity, respectively. Among the compounds in this series, goyazensolide (**5**) exhibited the strongest inhibitory effects on both proliferation and metabolic activity, whereas lychnopholide (**7**) displayed the most selective antiproliferative effects when compared to non-malignant HS-5 cells. Regarding their chemical structures, lychnopholide (**7**) differs from the other two sesquiterpenoids by a missing hydroxy group in the scaffold, while compounds **5** and **6** have the same scaffold but different moieties on the second hydroxy group, bearing either methacrylic acid (**5**) or angelic acid (**6**). Despite these structural similarities, different effects were observed in terms of the extent of inhibition or selectivity towards the malignant cell lines. Also, with respect to our previous studies, the above-mentioned compounds show superior effects and will therefore be the subject of further examinations.

## 5. Conclusions

In the present study, we isolated eight natural products, of which seven were investigated for their antiproliferative and antimetabolic effects on AML (HL-60 and MOLM-13) and MM (AMO-1 and KMS-12 PE) cell lines. All seven compounds dose-dependently exhibited activity against all tested cell lines, whereby sesquiterpenoids **5**, **6**, and **7** showed IC_50_ values in a very low micromolar range. Moreover, these effects were superior to the effects found against the non-malignant cell line HS-5, demonstrating, to some extent, selectivity towards malignant cells. With the promising results of this initial study, we will next focus on the compounds’ mechanism of action and thus further evaluate their usefulness as lead compounds in the treatment of AML and MM. Additionally, we once more demonstrated the potential of natural products in cancer therapy and confirmed their importance in the search for new therapeutics.

## Figures and Tables

**Figure 1 metabolites-15-00542-f001:**
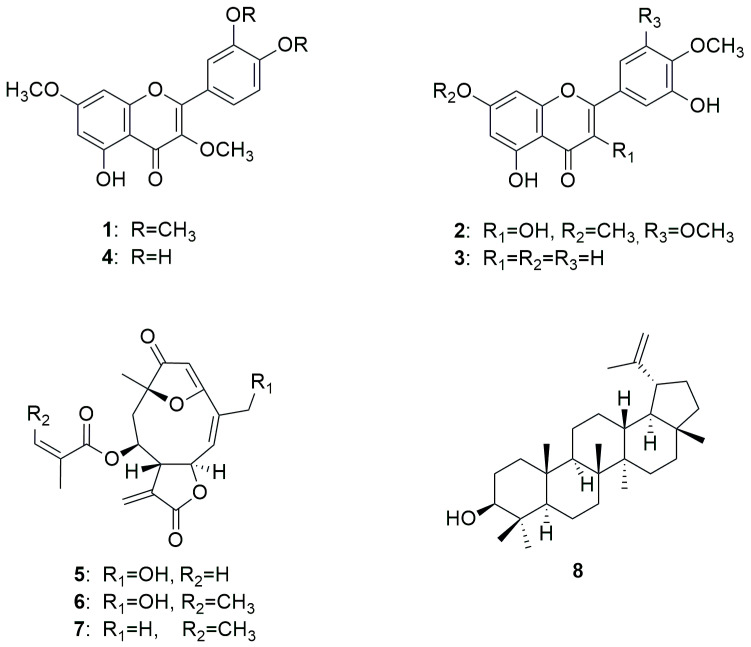
Chemical structures of isolated compounds from *L*. *ericoides*.

**Figure 2 metabolites-15-00542-f002:**
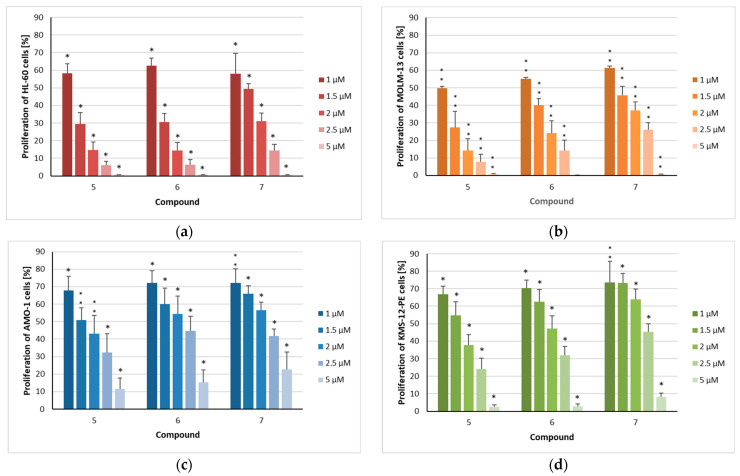
The proliferation of acute myeloid leukemia and myeloma cells after treatment with isolated compounds **5**–**7**: (**a**) HL-60, (**b**) MOLM-13, (**c**) AMO-1, and (**d**) KMS-12-PE cells were treated for 72 h with the indicated concentrations (1 µM/1.5 µM/2 µM/2.5 µM/5 µM) of the compounds. The mean proliferation + standard error (SE) of four to six independent experiments is depicted. Asterisks represent statistical significance (** *p* < 0.05 vs. control, * *p* < 0.001 vs. control).

**Figure 3 metabolites-15-00542-f003:**
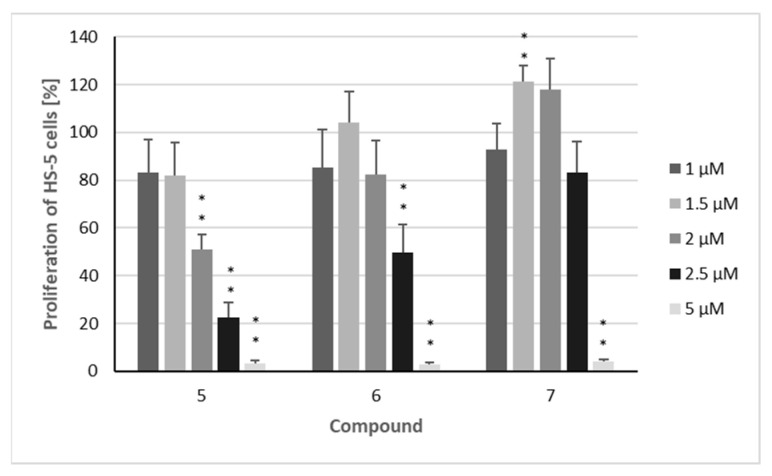
The proliferation of non-malignant HS-5 cells after treatment with isolated compounds. Percentage of cells proliferating after treatment with compounds **5**–**7**: HS-5 cells were treated for 72 h with the indicated concentrations (1 µM/1.5 µM/2 µM/2.5 µM/5 µM) of the compounds. The mean proliferation + standard error (SE) of four independent experiments is depicted. Asterisks represent statistical significance (** *p* < 0.05 vs. control).

**Figure 4 metabolites-15-00542-f004:**
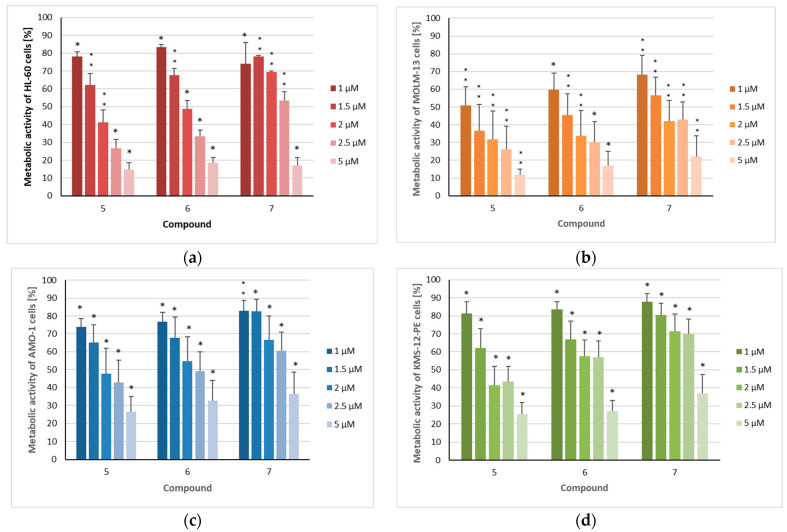
The metabolic activity of acute myeloid leukemia and myeloma cells after treatment with isolated compounds **5**–**7**: (**a**) HL-60, (**b**) MOLM-13, (**c**) AMO-1, and (**d**) KMS-12-PE cells were treated for 72 h with the indicated concentrations (1 µM/1.5 µM/2 µM/2.5 µM/5 µM) of the compounds. The mean metabolic + standard error (SE) of four to six independent experiments is depicted. Asterisks represent statistical significance (** *p* < 0.05 vs. control, * *p* < 0.001 vs. control).

**Figure 5 metabolites-15-00542-f005:**
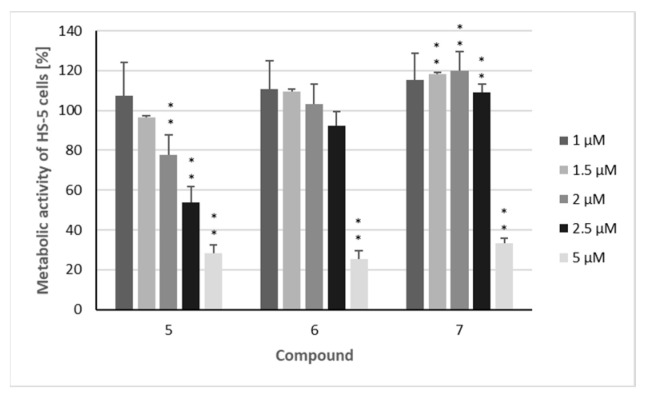
The metabolic activity of non-malignant HS-5 cells after treatment with isolated compounds. Percentage of cells metabolically active after treatment with compounds **5**–**7**: HS-5 cells were treated for 72 h with the indicated concentrations (1 µM/1.5 µM/2 µM/2.5 µM/5 µM) of the compounds. The mean metabolic + standard error (SE) of four independent experiments is depicted. Asterisks represent statistical significance (** *p* < 0.05 vs. control).

**Table 1 metabolites-15-00542-t001:** IC_50_ values, their 95% confidence intervals, and selectivity indices for compounds **5**–**7** for the inhibition of proliferation of AML (HL-60 and MOLM-13) and myeloma (AMO-1 and KMS-12 PE) cells. Results are given in µM and selectivity indices are shown in bold and italics.

Cell Line	Compound		
	5	6	7
**HL-60**	1.1 (1.0–1.2) µM, ***1.8***	1.2 (1.1–1.2) µM, ***2.1***	1.3 (0.9–1.5) µM, ***3.4***
**MOLM-13**	1.0 (0.7–1.2) µM, ***2.0***	1.1 (0.9–1.3) µM, ***2.2***	1.4 (1.1–1.6) µM, ***3.2***
**AMO-1**	1.6 (1.2–2.1) µM, ***1.2***	2.2 (1.6-n.d.) µM, ***1.1***	2.2 (1.7–3.0) µM, ***2.0***
**KMS-12-PE**	1.5 (1.3–1.7) µM, ***1.3***	1.8 (1.6–2.1) µM, ***1.4***	2.6 (n.d.) µM, ***1.7***
**HS-5**	2.0 (1.7–2.3) µM	2.5 (2.2–3.6) µM	4.3 (n.d.) µM

**Table 2 metabolites-15-00542-t002:** IC_50_ values, their 95% confidence intervals, and selectivity indices of compounds **5–7** for the inhibition of the metabolic activity of AML (HL-60 and MOLM-13) and myeloma (AMO-1 and KMS-12 PE) cells. Results are given in µM and selectivity indices are shown in bold and italics.

Cell Line	Compound		
	5	6	7
**HL-60**	1.7 (1.5–1.9) µM, ***1.5***	1.9 (1.8–2.1) µM, ***2.4***	2.9 (n.d.) µM, ***1.7***
**MOLM-13**	1.0 (n.d.-1.6) µM, ***2.6***	1.3 (n.d.-1.9) µM, ***3.5***	1.8 (0.9–2.9) µM, ***2.8***
**AMO-1**	2.0 (1.4–3.1) µM, ***1.3***	2.4 (1.5-n.d.) µM, ***1.9***	3.4 (2.3-n.d.) µM, ***1.4***
**KMS-12-PE**	1.9 (1.4–2.6) µM, ***1.4***	2.7 (2.1-n.d.) µM, ***1.6***	4.0 (n.d.) µM, ***1.2***
**HS-5**	2.6 (n.d.-3.7) µM	4.5 (n.d.) µM	4.9 (n.d.) µM

## Data Availability

The original contributions presented in this study are included in the article. Further inquiries can be directed to the corresponding author.

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
