# Peer review of "Flavonoids and Sesquiterpene Lactones from *Lychnophora ericoides* (Arnica-Do-Cerrado) and Their In Vitro Effects on Multiple Myeloma and Acute Myeloid Leukemia"

_metabolites, 2025, doi:10.3390/metabo15080542_

Round 1
Reviewer 1 Report
Comments and Suggestions for Authors
The article is interesting for natural product interest with promising bioactivities of isolated compounds from two different classes. However, some points can be improved, including the following notes:
- The article does not support some crucial data regarding the structure elucidation of the found compounds. Showing the NMR data, at least the δ (chemical shift) and other spectroscopic data of the work, will make the article more scientifically sound.
- Is there any solubility issue in the compound-containing test solution? The flavonoid derivative may potentially have different polarity and solubility in the used medium with that of terpenoid groups. As the preparation of the test solution is not available in the article, there is a lack of supportive statement to answer why the trend occurs. The purity of the isolated compound could also be considered as one of the factors for the bioassay results.
- As the work included statistical analyses, it is necessary to provide the standard error (SE) of any repeated measurements to observe the accuracy and precision of the values. We do not see the SE in the IC50 data while the SE are available in all inhibition assays.
Other specific comments can be found in the attached file.

Author Response
The article is interesting for natural product interest with promising bioactivities of isolated compounds from two different classes. However, some points can be improved, including the following notes:
- Many thanks for your nice words and the suggested improvements to our manuscript. Please find our answers below.
Comment 1: The article does not support some crucial data regarding the structure elucidation of the found compounds. Showing the NMR data, at least the δ (chemical shift) and other spectroscopic data of the work, will make the article more scientifically sound.
- We added the NMR data (chemical shifts and coupling constants) of all eight isolated compounds to the manuscript (Appendix A1).
Comment 2: Is there any solubility issue in the compound-containing test solution? The flavonoid derivative may potentially have different polarity and solubility in the used medium with that of terpenoid groups. As the preparation of the test solution is not available in the article, there is a lack of supportive statement to answer why the trend occurs. The purity of the isolated compound could also be considered as one of the factors for the bioassay results.
- The flavonoids and sesquiterpenoids of this study did not differ significantly in their polarities, which is why we isolated them from the same VLC fractions (A and B). They all showed good solubility in DMSO, which was used for the stock solution and showed no precipitation in the aqueous test media. In contrast, compound 8 was found only soluble in chloroform or hexane and therefore was excluded from the biological assays. As requested, we added the preparation of the stock solutions to the manuscript as well as purity of compounds, which we calculated using ERETIC. We found high purities for the sesquiterpenoids, but somewhat lower and more differing purities for the flavonoids. This could be a factor for the latter compounds, however, also the flavonoids with high purity did not exhibit pronounced effects. We added a statement on this in the discussion of the flavonoids’ activities (lines 316 to 318).
Comment 3: As the work included statistical analyses, it is necessary to provide the standard error (SE) of any repeated measurements to observe the accuracy and precision of the values. We do not see the SE in the IC50 data while the SE are available in all inhibition assays.
- The compounds exhibit efficacy within a very narrow concentration range (1–5 µM). As a result, the GraphPad Prism software was unable to calculate an IC₅₀ value based on a standard logarithmic dose–response curve. Nevertheless, the reliability of the data is clearly demonstrated in the figures. Additionally, GraphPad Prism 10 recommends reporting confidence intervals instead of the standard error of the mean. However, in some cases, no clear confidence interval could be determined. Therefore, confidence intervals are included in Table XX and Table YY where calculation was possible.
Comment 4: Other specific comments can be found in the attached file.
- We implemented the changes suggested in the attachment.
Reviewer 2 Report
Comments and Suggestions for Authors
Dear Authors,
Presented manuscript is well prepared.
In Introduction, I wolud suggest to add the numbers (1-7) of evaluated compounds (line 62). There is also a lack of an explanation why only 7, not all 8 compounds went thru biological evaluation.
In 4.2 paragraph:
line 313: The information about what was the ratio of of hexan to acetone used for maceration needs to be added.
line 323: The method/reason for selecting fractions A (line 323) and B (line 329) for further study should be determined.
Despite all used methods are well known, but the performed research is new in terms of the compounds tested and their potential use as a treatment in medicine.
Author Response
Dear Authors,
Presented manuscript is well prepared.
- Many thanks for your kind words and for honoring our work.
Comment 1: In Introduction, I wolud suggest to add the numbers (1-7) of evaluated compounds (line 62). There is also a lack of an explanation why only 7, not all 8 compounds went thru biological evaluation.
- We added the numbers to the introduction and an explanation for not testing compound 8, which is because of its very low polarity and the thus resulting low solubility in aqueous media. We added a statement on this in the results section (lines 186 to 187).
In 4.2 paragraph:
line 313: The information about what was the ratio of of hexan to acetone used for maceration needs to be added.
- This was a typing mistake. We did not use a mixture of hexane and acetone, but extracted sequentially with hexane, acetone, and methanol. We corrected this.
line 323: The method/reason for selecting fractions A (line 323) and B (line 329) for further study should be determined.
- As mentioned above and in the introduction (lines 58 to 61), we were focusing on sesquiterpenoids and methylated flavonoids. Fractions A and B showed UV and mass spectra that hinted at the compounds of our interest. We added this information in the materials and method section (now section 2.2, lines 111 to 112).
Despite all used methods are well known, but the performed research is new in terms of the compounds tested and their potential use as a treatment in medicine.
- Thank you again.
Round 2
Reviewer 1 Report
Comments and Suggestions for Authors
A revision on the article has been made addressing the comments given from the first-round review. Some noticeable changes can be seen for the addition of purity estimation of isolated compounds and NMR chemical shifts data, whereas no requested MS (mass spectra) is found.
A thoroughly double-checking on NMR-related symbols and chemical formula, especially for methoxy substituent, are strongly encouraged to this current version. The presentation must be in line with those provided in the IUPAC guideline.
The attached file is provided for details.

Author Response
Dear Reviewer,
Many thanks for the thorough review of our manuscript. As requested, we now added mass spectometric information of all isolated compounds to our manuscript, showing the measured m/z values and the calculated values for each signal. We also edited the presentation of the NMR data, now fulfilling the IUPAC guideline with respect to the writing of solvent names, symbols and chemical formulae. We also corrected the other formal issues raised by your side and marked all changes made this time in green colour (to be distinguished from the changes of the first revision, which are marked yellow). We hope that we revised our manuscript according to your expectations.
Kind regards,